# Biological Insights and Radiation–Immuno–Oncology Developments in Primary and Secondary Brain Tumors

**DOI:** 10.3390/cancers16112047

**Published:** 2024-05-28

**Authors:** Fabiana Gregucci, Kathryn Beal, Jonathan P. S. Knisely, Paul Pagnini, Alba Fiorentino, Elisabetta Bonzano, Claire I. Vanpouille-Box, Babacar Cisse, Susan C. Pannullo, Philip E. Stieg, Silvia C. Formenti

**Affiliations:** 1Department of Radiation Oncology, Weill Cornell Medicine, New York, NY 10065, USA; fgr4002@med.cornell.edu (F.G.); kab4027@med.cornell.edu (K.B.); jok9121@med.cornell.edu (J.P.S.K.); pgp4001@med.cornell.edu (P.P.); clv2002@med.cornell.edu (C.I.V.-B.); 2Department of Radiation Oncology, Miulli General Regional Hospital, Acquaviva delle Fonti, 70021 Bari, Italy; a.fiorentino@miulli.it; 3Department of Medicine and Surgery, LUM University, Casamassima, 70010 Bari, Italy; 4Department of Radiation Oncology, IRCCS San Matteo Polyclinic Foundation, 27100 Pavia, Italy; e.bonzano@smatteo.pv.it; 5Sandra and Edward Meyer Cancer Center, New York, NY 10065, USA; 6Department of Neurological Surgery, Weill Cornell Medicine, New York, NY 10065, USA; bac7002@med.cornell.edu (B.C.); scp2002@med.cornell.edu (S.C.P.); pes2008@med.cornell.edu (P.E.S.); 7Department of Biomedical Engineering, College of Engineering, Cornell University, Ithaca, NY 14850, USA; 8Department of Radiology, Weill Cornell Medicine, New York, NY 10065, USA

**Keywords:** radiotherapy, immunotherapy, brain metastases, glioblastoma, cancer biology, radioresistance, lipid, metabolism, fatty acids synthase, tumor microenvironment

## Abstract

**Simple Summary:**

Brain cancers, which can start in the brain or spread there from other parts of the body, are difficult to treat and often lead to severe health issues and death. Radiotherapy (RT) is a main treatment that helps control symptoms and can sometimes cure the disease, but many brain cancers resist it, especially those that start in the brain. Combining immunotherapy with RT has shown promise for treating cancers that spread to the brain, but has limited success with gliomas, the most common primary brain cancer. This review looks at why brain tumors resist RT, new strategies to overcome this, and the role of the tumor’s environment. We highlight key findings from recent research and identify new treatment opportunities to improve outcomes and survival rates for brain cancer patients.

**Abstract:**

Malignant central nervous system (CNS) cancers include a group of heterogeneous dis-eases characterized by a relative resistance to treatments and distinguished as either primary tumors arising in the CNS or secondary tumors that spread from other organs into the brain. Despite therapeutic efforts, they often cause significant mortality and morbidity across all ages. Radiotherapy (RT) remains the main treatment for brain cancers, improving associated symptoms, improving tumor control, and inducing a cure in some. However, the ultimate goal of cancer treatment, to improve a patient’s survival, remains elusive for many CNS cancers, especially primary tumors. Over the years, there have thus been many preclinical studies and clinical trials designed to identify and overcome mechanisms of resistance to improve outcomes after RT and other therapies. For example, immunotherapy delivered concurrent with RT, especially hypo-fractionated stereotactic RT, is synergistic and has revolutionized the clinical management and outcome of some brain tumors, in particular brain metastases (secondary brain tumors). However, its impact on gliomas, the most common primary malignant CNS tumors, remains limited. In this review, we provide an overview of radioresistance mechanisms, the emerging strategies to overcome radioresistance, the role of the tumor microenviroment (TME), and the selection of the most significant results of radiation–immuno–oncological investigations. We also identify novel therapeutic opportunities in primary and secondary brain tumors with the purpose of elucidating current knowledge and stimulating further research to improve tumor control and patients’ survival.

## 1. Introduction

Malignant central nervous system (CNS) tumors include a group of heterogeneous oncological diseases each defined by unique pathology, clinical presentation, site of involvement, and prognostic features. Many are also characterized by a relative resistance to treatments. Malignant brain tumors are distinguished as either primary tumors arising in the CNS, or secondary tumors that spread from other organs. Whether primary or secondary, brain tumors often cause significant morbidity and can have high mortality rates. In adults, the incidence of primary malignant brain tumors is approximately 7 per 100,000 individuals, of which approximately half are glioblastomas (GBM) [1]. Unfortunately, GBMs are very aggressive tumors with a median survival of approximately 12–18 months [2,3]. Secondary CNS cancers, brain metastases (BMs), are much more common than primary malignant brain tumors. In fact, approximately 20–40% of patients with solid tumors eventually develop BM [4], resulting every year in >200,000 BM patients in the USA. Non-small cell lung cancer, breast cancer, and melanoma are the three most common primary cancers that metastasize to the brain. Radiation therapy (RT) remains the primary treatment for BM, but subsets of solid tumor cancers are driven by specific mutations that can be treated effectively with targeted systemic therapies. Some of the specific pathways or receptors with mutations leading to solid tumors include epidermal growth factor receptor *EGFR*, *ALK*, *KRAS*, *ROS1*, *BRAF*, and *HER2*, which can be treated by drugs targeting those specific mutations. Many of the targeted drugs are small molecules which can cross the blood–brain barrier and thus are relatively effective in the upfront treatment of BM harboring these mutations [5,6,7]. However, in the absence of genetic targetable drivers, RT is typically used, sometimes in conjunction with surgery as the primary treatment. Many clinical studies and decades of experience have demonstrated the effectiveness of radiotherapy (RT) to control tumors and improve survival for both primary and secondary brain tumors. More recent studies have demonstrated that immunotherapy (IT) in conjunction with RT can be very effective for certain tumor types. This observation has revolutionized the landscape of brain cancer treatments [8,9]. As mentioned, the predominant role of RT in the treatment of primary and secondary brain tumors is well recognized [10,11], but the powerful and significant immunomodulatory capacity of RT has only recently been recognized [12]. The potential immunomodulatory effect of RT is more pronounced when high-dose RT such as radiosurgery (SRS) or hypo-fractionated stereotactic RT (HFSRT) is used. Initial results of both preclinical and clinical studies of HFSRT and IT are promising, but the optimal combination and sequence of these therapeutic approaches remain under investigation.

In this context, research is ongoing to elucidate the interactions between tumor cells, the tumor microenvironment (TME), and the host’s innate/adaptive immune system. Understanding how interactions modulate and influence cancer development, growth, and response or resistance to treatment is obviously important to improve the therapeutic index and cure more brain cancers.

Based on this background, we provide an overview of radioresistance mechanisms, strategies to overcome radioresistance, the role of TME, and the main results of radiation–immuno–oncological investigations as well as novel tailored therapeutic opportunities in primary and secondary brain cancers. The goal of this review is to elucidate current knowledge and stimulate further research to improve tumor control and patients’ survival Several other immunotherapeutic approaches are under investigation, including peptide, multi-peptide, and dendritic cell-based vaccines, oncolytic viral therapy, antibody drug conjugates (ACPs), and CAR-T cell therapy (Figure 1).

## 2. Radioresistance

Radioresistance in cancer cells is defined as the cellular capacity of resisting the lethal effects of radiation, resulting in tumor recurrence. Understanding radioresistance pathways may help identify potential strategies to overcome radioresistance in brain tumors.

Fortunately, the ablative doses of SRS and HFSRT are relatively effective for most BM. However, there are a few types of BM that are relatively radioresistant. These include BM from melanoma, renal cell carcinoma, sarcoma, and gastrointestinal cancers [13,14]. Primary malignant brain tumors [15] are often initially responsive to radiation, but the high-grade tumors which represent the majority of primary brain tumors eventually recur despite high initial doses of RT. Radioresistance is a complex phenomenon that involves multiple pathways, including DNA repair, apoptosis, and inflammation. One of the main mechanisms of radioresistance is the activation of DNA damage repair pathways [16]. After exposure to radiation, cells activate various DNA repair pathways, such as homologous recombination, non-homologous end-joining, and base excision repair, to repair the DNA damage induced through RT. The activation of these pathways promotes cell survival and results in radioresistance. Another pathway that plays a role in radioresistance is apoptosis or programmed cell death. Radiation can induce apoptosis in tumor cells, but radioresistant tumors can evade apoptosis by upregulating anti-apoptotic proteins such as Bcl-2 and Bcl-xL [17,18]. These proteins inhibit the intrinsic apoptotic pathway through blocking the release of cytochrome c from the mitochondria and activating survival pathways such as PI3K/Akt. Inflammation also can result in radioresistance. Radiation-induced inflammatory response can contribute to tumor progression and treatment resistance, as inflammation can activate survival pathways, promote angiogenesis, and recruit immune cells that can suppress the immune response against the tumor [19,20].

## 3. Radiation Optimization Strategies

Some strategies developed and under investigation to overcome tumor resistance include SRS or HFSRT ± IT, innovative approaches for radiation delivery in terms of temporal and spatial fractionation, particle RT, and tailored systemic therapy.

SRS or HFSRT is non-invasive, high-dose, and highly conformal, typically used to treat brain metastases, occasionally primary malignant tumors such as GBM, and benign tumors such as acoustic neuromas. Because of the conformality of these treatment techniques, SRS or HFSRT minimizes exposure to surrounding normal healthy tissue while delivering a focused high dose of radiation to a specific target or tumor. RT at a high dose per fraction can overcome radioresistance, as opposed to RT at a lower dose per fraction, as a high dose per fraction induces more direct DNA damage and indirect cellular injury due to the ablative effect on the tumor vasculature [21]. Moreover, preclinical studies have shown that SRS or HFSRT can stimulate antitumor immune responses by inducing immunogenic cell response, promoting antigen presentation, and enhancing infiltration of immune cells into the TME [8,9]. IT such as immune checkpoint inhibitors or adoptive T-cell therapy can further enhance these effects by removing immunosuppressive barriers and activating tumor-specific T cells [22,23,24,25]. In this direction, heightened application of technical and biological knowledge allows innovative approaches for radiation delivery that could elicit an in situ vaccination effect from RT in combination with checkpoint blockade, increasing diversity among tumor-infiltrated lymphocytes and stimulating recognition of tumor mutation-associated neoantigens [26,27].

Recent and contemporary areas of exploration in RT application include FLASH-RT and PULSAR (personalized ultrafractionated stereotactic adaptive radiotherapy) which represent examples of different temporal fractionation RT delivery. FLASH-RT delivers radiation at ultra-high dose rates with specific beam parameters able to effectively treat tumors without inducing adverse toxicity within the surrounding normal tissues [28]. The physical, chemical, and biological mechanisms of FLASH are unknown, but several studies are exploring these [29]. PULSAR is characterized by the possibility of splitting the course of RT in time by weeks or months, through giving a large radiation dose with each fraction. This administration modality aims to improve the tolerance of organs at risk and to adapt the treatment based on tumor response and modification of its TME [30]. In terms of alternative spatial fractionation, RT approaches such as GRID, lattice RT, and minibeam RT are under investigation [30,31,32,33]. GRID and lattice RT are types of high-dose spatially fractionated radiation therapy, based on 2D and 3D techniques, respectively, that allow ablative doses to be delivered to large lesions without increasing toxicity, with alternating high-dose and low-dose areas as peaks and valleys [31,32]. Minibeam provides preclinical ultrahigh dose rates RT of the order of up to 100 Gy/s or higher, using a strong spatial modulation of the dose, as the irradiation is performed with arrays of narrow parallel beams (0.5–1 mm) spaced 1.5 to 4 mm apart, resulting in a highly heterogeneous dose distribution [31,33].

Clinical trials are ongoing to investigate the safety and efficacy of combining different RT approaches with IT in patients with both primary brain tumors and BM. While initial results have been promising, more data are needed to determine the optimal timing, sequencing, and dosing of these treatments [34].

In addition to developing more efficient combination therapies, precise RT targeting of the tumor is essential for minimizing side effects and improving the efficacy of SRS. Advanced imaging techniques such as MRI and PET, as well as innovative technologies such as the Gamma Knife, Cyberknife, Novalis, and ZAP-X Gyroscopic SRS devices have been developed to improve the precision of SRS, increasing the potential efficacy and reducing the risk of side effects.

Particle therapy with protons and heavier charged particles is a modality of RT that can be used to treat radioresistant tumors. Compared with conventional photon RT, particle therapy is characterized by high-linear energy transfer (LET) radiation that theoretically leads to higher biological effectiveness (RBE) as it can cause more DNA double-strand breaks than comparable photon RT [35]. Practically, though, there is no obvious clinical evidence of the advantages of particle therapy in terms of overcoming radioresistance. The most commonly used particle therapy, proton therapy, is most recognized for its physically advantageous feature of the Bragg peak. The Bragg peak imparted by proton beams enables a rapid drop-off of dose beyond the depth of the targeted tumor, which reduces exposure of normal tissue and is particularly useful for larger field treatments or pediatric patients.

Beyond the physical advantages of proton therapy, there are biologic strategies to improve the radiosensitivity of tumors, such as combining RT with drugs that enhance biologic sensitivities. One example is combining RT with PARP inhibitors that target DNA repair pathways. This combination has shown promise in preclinical studies for enhancing the sensitivity of brain tumors to radiation [36]. Similarly, targeting anti-apoptotic proteins such as Bcl-2 and Bcl-xL can sensitize radioresistant tumors to radiation [37]. In addition, inhibiting inflammatory pathways such as the NF-kB pathway has been shown to sensitize tumors to radiation and reduce treatment resistance [38].

Overall, the strategies described above have been promising in preclinical and clinical studies for enhancing the sensitivity of brain tumors to radiation and overcoming radioresistance.

## 4. The Role of Tumor Microenvironment in Brain Cancers

Over the past several years, it has become clear that the initiation, growth, progression, and recurrence of primary and secondary brain tumors rely heavily on regulatory signals and factors that emanate from the TME and factors that tumor cells direct at constituents of the TME. From the time of tumor initiation to recurrence after treatment, there are complex and constant interactions between tumor and non-tumor host cells. Tumor cells express ligands and receptors through which they interact with the extracellular matrix and cells that reside in or home to the microenvironment [39,40]. The interaction between neoplastic and non-neoplastic cells typically involves downstream activation of transcriptional factors that either activate or inhibit a set of target genes that mediate phenotypic and/or functional changes in cells in the microenvironment [39,40]. Tumor cells also release factors that antagonize the anti-tumor immunity activities of various resident or homing immune cells that are poised to mount anti-tumor immunity. Microglia, the resident macrophages of the CNS, tumor-associated macrophages (TAMs), myeloid-derived suppressor cells (MDSCs), B, T, and NK cells constitute some of the key immune cells in the microenvironment of brain tumors [41,42]. The outcome of anti-tumor and pro-tumorigenic interactions of host cells with neoplastic cells ultimately defines a microenvironment that is either favorable or detrimental to tumor growth, progression, and/or recurrence. Astrocytes also constitute key components of the tumor microenvironment in the brain. Astrocytes are the most abundant cells in the brain, where they play various roles in health and disease [43,44,45]. In the context of brain tumors, astrocytes, in addition to possibly being cells of origin for many primary tumors, can become reactive and impact disease progression. For instance, tumor-associated astrocytes have been shown to support tumor cell proliferation and migration [44] as well as anti-inflammatory responses. Therefore, a thorough understanding of the formation of brain tumors, their mechanisms of progression, and reasons for recurrence after treatment requires characterization of the molecular and genetic factors that mediate and regulate the temporal and spatial roles of non-neoplastic cells, especially immune cells [46,47]. This ultimately may lead to the development of stroma-directed glioma therapies that can be combined with anti-neoplastic cell-targeted therapies for more effective treatment of primary and secondary brain tumors.

## 5. Radiotherapy as Standard Care in Primary Brain Tumor and Brain Metastasis

Radiotherapy (RT) has been used to treat cancer since the early 20th century. At the end of the 19th century, in 1896, Wilhelm Conrad Roentgen, a German professor of physics, presented a lecture describing the “X-ray”. Systems were rapidly developed to use X-rays for diagnostic purposes and then, within a few years, X-rays were used to treat cancer. It was quickly recognized that fractionated, small, daily doses of X-rays delivered over an interval of several weeks enabled optimal recovery of the normal tissues in the radiation field while offering patients local tumor control that in some cases could result in a cure. Decades later, progress in technology has perfected dose delivery with optimal sparing of normal tissues, often enabling the use of hypofractionation.

After the mid 1930’s, X-ray therapy was mainly conducted with electron accelerating machines developed primarily in the United States and England. Some of the earliest studies were directed towards radiosensitive tumors such as lymphomas and some brain tumors such as oligodendrogliomas. Ballard et al. [48] and Wagner et al. [49] published their outcomes in relatively modern series that included patients treated in the 1940’s and showed dramatic improvement in 5- and 10-year survival in patients who received radiation therapy (RT) following surgery compared with surgery alone.

RT has also long been used for the treatment of BM. For several decades, the standard of care for patients with BM was whole-brain radiation therapy (WBRT), typically delivered in 10 fractions of 300 cGy each to a total dose of 3000 cGy (30 Gy) [50]. A practice-changing study by Patchell et al. [51] demonstrated the role of post-operative WBRT after surgical resection of BM [51]. Although WBRT is effective in controlling BM, it unfortunately has significant toxicity in many patients. It is well recognized that WBRT can impact memory and induce dementia in some patients [52,53]. Additionally, the 2 weeks of WBRT often delay initiation of systemic therapies. Furthermore, WBRT negatively impacts the immune system as it affects circulating lymphocyte counts [54]. Multiple publications in the past 20 years demonstrate that lymphocytopenia following RT is associated with poor outcomes.

Finally, since WBRT can induce tumor-associated edema that can exacerbate neurological symptoms, corticosteroids are typically used, with their associated immunosuppressive effects representing a double-edged sword as they are often necessary to help control significant neurologic symptoms until the tumor is well treated.

Fortunately, the introduction of SRS has revolutionized the management of BM over the past several decades [4,11]. SRS is commonly used in cancer centers throughout the USA, Europe, and Asia to treat BM as well as other types of brain tumors (primarily benign) and to ablate vascular malformations. In the past, SRS was often offered only to patients with one to three BMs. However, relatively recently, SRS technology has advanced such that we are often able to treat patients with several or many (more than 10) BMs and avoid the toxicity and delays associated with WBRT [55,56]. In addition, SRS has the distinct advantage of delivering an ablative dose of radiation in a highly accurate manner with a dramatic drop-off of dose outside the targeted tumor, which has been shown in randomized trials to provide better neurocognitive outcomes than WBRT. It is best suited for very distinct tumors, such as BMs that are well defined and relatively small in volume. For slightly larger BMs, HFSRT is used to deliver an ablative dose of RT, usually divided into 3 or 5 treatments. It was also recently demonstrated that high-dose RT, such as SRS or HFSRT, induces an immunogenic response [8].

Likewise, RT is a cornerstone in the treatment of GBM. In 2005, the results of a randomized phase III trial [10] testing the role of adjuvant RT (60 Gy in 30 fractions) with concomitant and adjuvant temozolomide (TMZ) after maximal surgical resection compared with RT alone showed a median survival of 14.6 months with RT + TMZ vs. 12.1 months with RT alone and a 2-year survival rate of 26.5% with RT + TMZ vs. 10.4% with RT alone, changing the practice of GBM management.

Over the years, mostly in patients with poor prognosis, hypofractionated RT (10–15 fractions) and HFSRT (3–5 fractions) schemes ± TMZ have been tested, with the purpose of reducing overall treatment time and offering patients possibly effective and safe palliative care [3]. In 2004, Roa et al. published the results of non-inferiority trial comparing 40 Gy/15 fractions vs. 60 Gy/30 fractions in patients 60 years and older, showing no difference in survival between patients receiving standard RT or short-course RT [57]. Additional studies also demonstrated the safety and efficacy of that shorted RT regimen with concurrent and adjuvant temozolomide in elderly and infirm patients [58]. A phase II clinical trial also demonstrated the efficacy and tolerability of HFSRT (6 Gy × 6) and concurrent TMZ and bevacizumab with impressive (but not significant) prolonged overall survival in patients whose tumors did not harbor MGMT methylation, thus perhaps suggesting a heightened immune response in these patients when treated with an immunogenic fractionation scheme [59].

More progress is warranted to treat both high-grade gliomas and BM. Tailored strategies to overcome treatment resistance show promise. Particularly, how to elicit immunogenicity and recruit the host’s immune system to combat brain cancers has grown as a central area of investigation.

## 6. IT-RT in Brain Metastasis

Ipilimumab was the first checkpoint inhibitor approved for the use of cancer treatment after Hodi et al. demonstrated a survival advantage over standard chemotherapy in patients with advanced melanoma [60]. Patients with BM from melanoma were excluded from this study, given their notoriously bad outcomes with a median survival rate of only a few months. However, following the encouraging results of patients with systemic metastatic disease from melanoma, patients with BM were empirically treated with SRS and ipilimumab. The results were astounding with many patients achieving durable control and indeed, even cure in some [61,62]. Those studies showed that there was prolonged survival in many of these patients treated with SRS and ipilimumab, with results clearly superior to historic controls. Around that time, preclinical evidence was published demonstrating the immunologic effect of hypo-fractionated radiation therapy with clear infiltration of TILs after doses similar to what was and is used for larger BMs [63]. Additionally, a preclinical study showed a dramatic survival advantage and abscopal effect in mice with bilateral flank tumors treated with RT compared with those with only one tumor treated with concurrent checkpoint inhibitors. These lessons were rationally applied to our practice, as patients with melanoma BM in particular had very dire outcomes otherwise. Beyond our experience with ipilimumab and SRS, we recognized that other checkpoint inhibitors also induced a dramatic response with SRS [64]. In 2017, Anderson et al. [65] published their experience of pembrolizumab and SRS, demonstrating a remarkable and rapid response of BMs in comparison to ipilimumab and SRS or SRS alone. Furthermore, there were reports of high-dose hypofractionated RT for body metastases delivered concurrent with ipilimumab that induced an abscopal effect, clearly demonstrating the immunogenicity of high-dose hypofractionated RT [66,67].

Since these early studies, there have been many publications demonstrating similar effects and variable radiographic responses with perhaps heightened “adverse radiation effects” or even radiation necrosis, following IT and SRS for BM. Interestingly, these observed effects seem to correlate with good outcomes, similar to adverse immune effects seen systemically after IT alone. Importantly, at ASTRO 2022, the Memorial Sloan Kettering Cancer Center (MSKCC) presented their experience of patients who had survived >5 years following SRS and IT for melanoma BM. In this group of patients, >25% eventually had their BMs resected due to worrisome changes, and none of those specimens had viable disease. In fact, all of them comprised reactive tissue and showed evidence of gliosis, indicating a very robust treatment response unlike responses seen in BM treated with RT alone.

Finally, IT in the form of checkpoint inhibitors is nowadays applied to many immunogenic cancers, such as lung and renal cell cancers. In fact, a clinical trial at MSKCC is exploring the use of checkpoint inhibitors in combination with VEGF inhibitors and SRS for renal cell cancer BM. This trial takes advantage of the anti-edema properties of VEGF inhibition that allow avoidance of (immunosuppressive) corticosteroids in patients with BM. This combination and avoidance of corticosteroids may yet heighten the effect of concurrent IT and SRS.

## 7. IT-RT in High Grade Gliomas

Immune checkpoint inhibitors have demonstrated only modest activity when used as part of the management of patients with GBM, either at recurrence or as part of initial therapy. The phase II/III trial NRG-BN007 (NCT04396860) [68] compared the usual treatment of RT and temozolomide to RT in combination with IT (ipilimumab and nivolumab) for treating patients with newly diagnosed unmethylated MGMT (tumor O-6-methylguanine DNA methyltransferase) GBM. Recently, a futility analysis led to the early closure of that trial because the investigational arm did not meet protocol-defined criteria for progression-free survival [69].

Another trial, CheckMate 143, compared the efficacy and safety of nivolumab administered alone versus bevacizumab in patients diagnosed with recurrent GBM. It also further evaluated the safety and tolerability of nivolumab administered alone or in combination with ipilimumab in patients at different disease points of their recurrent GBM [69,70]. In total, 439 patients with recurrent GBM were enrolled at their first failure after standard partial brain RT and temozolomide, and 369 were randomized on the trial. The study was negative overall, but in subset analyses, patients with methylated MGMT promoters had a survival advantage with nivolumab and temozolomide, with a median overall survival of 33.4 months vs. 16.9 months for patients who lacked MGMT promoter methylation [70,71].

It appears that unlike in other cancers, programmed death ligand 1 (PD-L1) levels do not predict response to checkpoint inhibitors in GBM [72]. However, these and other discouraging findings have not shut down investigator interest in trying to improve control of malignant gliomas through employing immune checkpoint inhibitors.

The futility of the trials conducted so far has been disappointing, but there may be factors related to current management approaches that can be ameliorated or avoided in subsequent trials. Lymphopenia caused by temozolomide and corticosteroids may blunt potential immune responses to checkpoint inhibitors. Radiotherapy also contributes to lymphopenia, and the current standard prescription of 2.0 Gy doses administered over 30 daily fractions is less immunogenic than a hypofractionated approach with high doses of RT with each treatment, which is known to be immunogenic. However, the standard treatment approach to GBM of 30 daily fractions of RT with concurrent temozolomide was derived through decades of clinical trials demonstrating superior survival to other competing approaches. These approaches were adopted before the current understanding and recognition that high-dose hypofractionated RT was immunogenic and may be very effective in tandem with IO (as with some types of BM).

Some factors inherent to GBM are likely to continue to be refractory. GBM cells are spatially heterogeneous. Clonal evolution occurs within a tumor under differing microenvironments, and this results in genotypically and phenotypically different subclones that may differ significantly in their response to therapy [73]. The selective pressure of tumor therapies also exerts an influence—more aggressive and resistant clones proliferate through treatment.

Tumors promote an immunosuppressive microenvironment within a relatively immune-privileged organ. Soluble factors produced by the tumor, such as kynurenine and adenosine, initiate signal cascades that inhibit the immune system [74,75] and tryptophan secretion causes paracrine sequestration of T cells in bone marrow [76]. T cells found within GBM are both sparse and ineffective, exhibiting anergy, tolerance, and exhaustion from persistent exposure to tumor-associated antigens in the presence of inhibitory receptors. This lowers IL-2 production and blunts antitumor immune responses. Also, despite the presence of high levels of tumor-associated macrophages (both brain-resident microglia and bone marrow-derived macrophages), they are largely inactive and play a supportive role in the growth of tumor cells.

A phase II trial published by Omuro et al. tested some of the treatment-related factors mentioned above through giving HFSRT (6.0 Gy × 6 to resection beds and unresected solid tumor and 4.0 Gy × 6 to FLAIR abnormality) with concurrent and adjuvant temozolomide and with concurrent bevacizumab for prophylaxis against RT injury and to decrease tumor-associated edema (and avoid corticosteroids). The regimen was well tolerated; indeed, the demonstrated median OS was 19 months for all patients, with an OS of 22 months in the subset of patients whose tumors did not demonstrate MGMT methylation. This result was not statistically significant, but is nonetheless thought-provoking as MGMT-unmethylated tumors are notoriously aggressive and may thus be more immunogenic and more likely to benefit from a more immunogenic treatment strategy such as HFSRT [58].

Several other immunotherapeutic approaches are under investigation, including peptide, multi-peptide, and dendritic cell-based vaccines, oncolytic viral therapy, antibody–drug conjugates (ACPs), and CAR-T cell therapy. However, the results remain immature and require further clinical evaluation [77].

## 8. A New Strategy of Tailored Therapy for Radioresistant Brain Tumors

Targeting fatty acids (FAs), a class of lipids, is emerging as a promising approach against brain tumors [78]. FAs are long-chain hydrocarbons that can be classified into saturated fatty acids (SFAs), mono-unsaturated fatty acids (MUFAs), and poly-unsaturated fatty acids (PUFAs), based on the carbon lengths and degrees of desaturation. They serve as building blocks of many lipids and have shown important roles in energy storage, insulation, and cell signaling. In the brain, FAs are critical for numerous processes, including (but not limited to) neurotransmitter trafficking [79] and myelination [80]. Importantly, while glucose is the predominant energy substrate in the brain, long-chain FAs can be catabolized as an alternative source to produce ATP [81]. In the context of a tumor, brain cancer cells overexpress the fatty acid synthase (FASN) to synthesize new FAs to sustain their growth [82,83]. Specifically, they preferentially utilize short-chain FAs to facilitate its absorption and acquire a growth advantage when compared to normal brain cells [78,84]. Supporting the critical role of FASN in the carcinogenesis of brain tumors, a recent phase II clinical study of relapsed high-grade astrocytoma showed an objective response rate of 56%, complete response rate of 17%, plus partial response rate of 39% in patients receiving FASN inhibition in combination with bevacizumab [85]. Progression-free survival at 6 months (PFS6) with the combination of FASN inhibition and bevacizumab was 31.4%, which was statistically significant when compared with historical controls of bevacizumab as monotherapy (BELOB trial; PFS6: 16%) [86].

The pro-tumorigenic role of FA metabolism extends beyond neoplastic development. Indeed, accumulating evidence reveals that resistant cancers, including brain tumors, reprogram their energy metabolism to survive anti-cancer treatment such as radiation therapy (RT) [87,88,89]. Such metabolic rewiring is a consequence of excessive levels of reactive oxygen species (ROS) that impair cellular homeostasis [90,91,92,93]. Therefore, to decrease ROS toxicity and maintain cell survival, irradiated cancer cells reprogram their metabolism to elicit a cytoprotective response to oxidative stress, in part through increasing FA metabolism [93,94,95]. FASN-mediated de novo FA synthesis has been linked to RT resistance and poor outcomes [96,97,98,99], giving rise to FASN blockade strategies to sensitize tumors to RT [97].

We recently showed that RT promotes FASN-mediated unsaturated fatty acids to protect GBM cells from undergoing apoptosis and sustain their survival, thus further reinforcing the rationale for targeting FASN in irradiated brain tumors [95]. However, a recent study demonstrated that supplementation with exogeneous MUFAs and di-unsaturated FAs radio-sensitized cervical cancer via a p53/CD36-dependent mechanism [100]. These potential discrepancies might be explained by the activation of two distinct pathways that permit the accumulation of FAs in cancer cells, namely FASN-mediated de novo FA synthesis and CD36-mediated FA uptake. Additional work is warranted to better characterize the function of SFAs, MUFAs, and PUFAs in radiation response with respect to tumor metabolic landscape at baseline and the pathway responsible for FA accumulation in cancer cells after RT.

Aside from its role in brain tumor development and RT resistance, FA metabolism was also recently reported to have an immunosuppressive role in irradiated GBM. Specifically, Jiang et al. demonstrated that fatty acid oxidation (FAO; also referred as fatty acid β-oxidation) upregulated the “don’t eat me signal” CD47 in GBM to hinder the phagocytosis properties of macrophages [101]. These data suggest that targeting FAO with an inhibitor of carnitine palmitotransferase-1 (CPT-1) to prevent the transport of FAs into mitochondria for oxidation is a potential new approach to improve the combination of focal RT with anti-CD47 antibody to stimulate immunity against brain tumors.

While emerging evidence points towards a major role of FA metabolism in mediating brain tumor carcinogenesis, resistance to treatment, and immunosuppression, additional work is warranted to successfully translate the combination of RT and FA metabolism to target agents to control brain tumors (Figure 2).

## 9. Conclusions

Primary and secondary brain tumors are a group of heterogeneous oncological diseases that, despite therapeutic efforts, often cause significant mortality and morbidity across all ages. Historically, RT has played a key role in the treatment of brain cancers, delivered for the purpose of improving local control and/or palliating symptoms. However, the definitive goal of cancer treatments, to cure patients, remains elusive for many brain cancers; thus, translational research by clinicians and scientists continues in earnest around the world. Over the years, several preclinical studies and clinical trials have been designed to elucidate possible mechanisms of resistance and treatment paths more effective than standard care. IT in synergistic association with RT, mostly HFSRT, has revolutionized the clinical management of many malignancies, including brain metastases. Single or combination immune checkpoint inhibitors, such as anti-cytotoxic T-lymphocyte associated protein 4 (CTLA-4) and anti-programmed cell death 1 (PD-1), associated with stereotactic RT showed remarkable and durable responses in patients with BM from melanoma without severe side effects. Moreover, an abscopal effect was observed in some cases, highlighting the immunogenic effect of IT-RT combination therapy. These emerging data support the same therapeutic application in many other immunogenic cancers, such as lung cancer BM. Nevertheless, the impact of IT-RT on GBM and resistant metastatic tumors is still under investigation. Some evidence derived through clinical trials suggests that methylated MGMT GBM may have a therapeutic advantage, based on adding nivolumab to standard therapy; on the other hand, in unmethylated MGMT GBM, HFSRT could be used to induce an immunogenic reaction that improves survival. In this scenario, investigating radiation and drug resistance mechanisms presents a major challenge to the treatment of primary and secondary brain tumors.

## Figures and Tables

**Figure 1 cancers-16-02047-f001:**
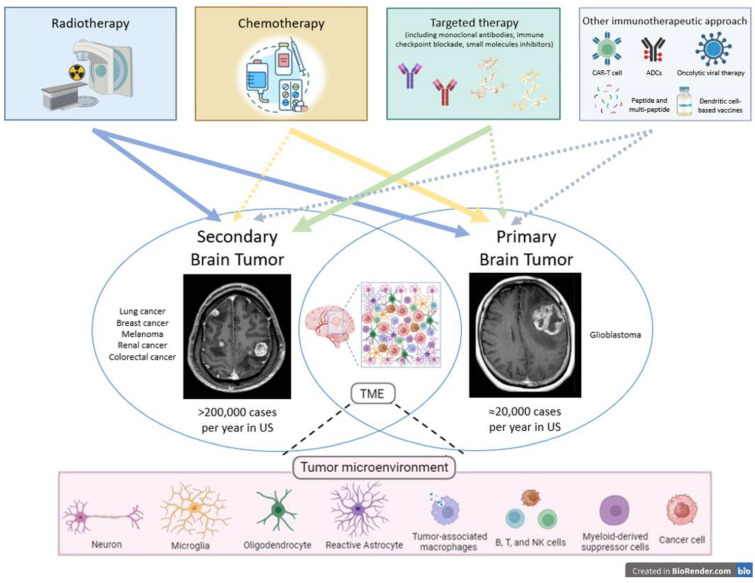
The tumor microenvironment of primary and secondary brain tumors refers to the complex network of cells, molecules, and structures surrounding the tumor cells within the brain. This environment plays a critical role in influencing different aspects of tumor growth, invasiveness, and treatment response. Between oncological treatments for both primary and secondary brain tumors (solid arrows denote significant treatment impacts, while dashed arrows signify less significant therapeutic results), radiotherapy plays a pivotal and fundamental role in both contexts. Concerning systemic treatments, chemotherapy remains a cornerstone in the treatment of primary brain tumors, whereas in secondary brain tumors, targeted therapies, including monoclonal antibodies, immune checkpoint blockade, and small molecules with inhibitor functions, have revolutionized the landscape of brain cancer treatments. Several immunotherapeutic approaches are currently under preclinical and clinical development.

**Figure 2 cancers-16-02047-f002:**
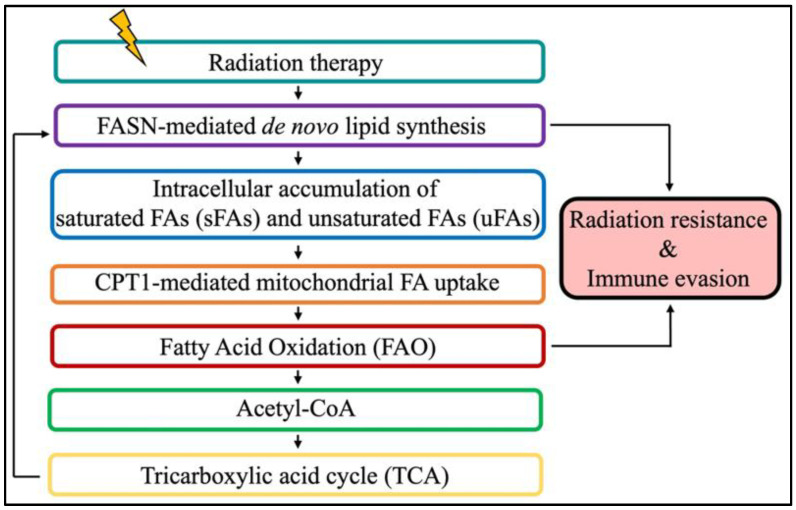
Radiation therapy (RT) promotes radiation resistance and immune evasion by reprogramming the tumor metabolism. Briefly, RT induced de novo lipid metabolism mediated by fatty acid synthase (FASN) to accumulate intracellular fatty acids (FAs) that can be used as an energy supply for fatty acid oxidation (FAO) in the mitochondria. Acetyl-CoA, the end product of FAO, can enter the tricarboxylic acid cycle (TCA cycle, also known as the Krebs cycle) to produce citrate. Citrate is then exported into the cytosol as substrate for de novo lipogenesis to subsequently promote radiation resistance and immune evasion.

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
