# Peer review of "Biological Insights and Radiation–Immuno–Oncology Developments in Primary and Secondary Brain Tumors"

_cancers, 2024, doi:10.3390/cancers16112047_

Round 1

Reviewer 1 Report

Comments and Suggestions for Authors

This is a well-structured and written review study where the authors include neoplasms of the central nervous system where the problem with resistance to chemotherapy and radiotherapy is addressed in different molecular mechanisms, to identify the resistance mechanisms for improved outcomes after RT and other therapies. Some comments are listed below:

1. correct punctuation errors, standardize the use of abbreviations or complete words, and correct writing of proteins (not italics) and genes (italics), some abbreviations are not defined, and define the abbreviation only once and then refer to it.

2. The paragraphs that contained lines 166-186, 230-238, 321-342, 356-377, and 443-448, have no scientific basis. Include references

3. homogenize the writing of references according to the journal's instructions

Author Response

Dear reviewer, thanks for your suggestion that improve the manuscript quality.

  1. The paper was reviewed to correct punctuation, abbreviation and writing.
  2. Where possible, the references were added
  3. The reference were reviewed according with Journal police

Reviewer 2 Report

Comments and Suggestions for Authors

In the present work Gregucci et al., reviewed the literature in order to provide an overview on the radioresistance mechanisms, strategies to overcome radioresistance, the role of the tumor microenvironement, and the main results of radiation-immuno-oncology investigations as well as novel tailored therapeutic opportunities in primary and secondary brain cancers. The goal of their review was to elucidate the current knowledge on these topics and stimulate further research to improve tumor control and patients’ survival. The authors have presented very well and in a way that it is easy to understand.

The present work is very interesting and the topic covered is of outmost importance. The authors have done a good job in describing the main aspects of radiation therapy, immunotherapy and their combinations.

One observation for the present work would be to supplement their work with a table summarizing their findings. It would be easier to have a table showing the main aspects of the present review. In other words, showing the advances in RT and IT as well as their combinations.

Something that would have been of interest would be the personalized approaches in brain tumor treatments. In particular, how do personalized treatments (for example, mutation-specific inhibitors) can affect, enhance other forms of treatment?

Comments on the Quality of English Language

minor editing

Author Response

Dear reviewer,

Thank you for your thoughtful comments and suggestions on our work. We appreciate your recognition of the importance of the topic and your positive feedback on our description of the key aspects of radiation therapy, immunotherapy, and their combinations.

Regarding your suggestion to include a table summarizing our findings, we acknowledge that such a table could indeed facilitate a quick overview of the advances in radiation therapy (RT), immunotherapy (IT), and their combinations. However, our paper predominantly focuses on the intricate mechanisms that govern the immuno-radiobiological response in primary and secondary brain tumors. We aim to provide an in-depth understanding of the interactions within the tumor microenvironment that contribute to radioresistance and how these mechanisms can be overcome with various approaches.

Given the vastness, heterogeneity, and complexity of the topics covered, we believe that a table might not effectively capture the nuanced interactions and could potentially be misleading or confusing. Our goal is to offer a comprehensive and detailed exploration of these complex mechanisms, which we feel is better conveyed through the detailed narrative provided in the paper.

As for the personalized approaches in brain tumor treatments, we agree that this is an area of significant interest. The impact of mutation-specific inhibitors and other personalized treatments on enhancing other forms of therapy is indeed crucial. While this paper focuses more on the general mechanisms and potential strategies to overcome radioresistance, we acknowledge that personalized medicine represents a promising frontier that warrants detailed exploration in future work.

Thank you again for your valuable feedback. We hope our explanations clarify the focus of our paper and the rationale behind our decision regarding the presentation of the information.

Reviewer 3 Report

Comments and Suggestions for Authors

Gist/Summary:  The authors in this enlightening review discuss the role of primary and secondary brain tumors in lieu of their therapeutic efforts, mortality and morbidity.  They focus on radiotherapy (RT) as a key treatment parameter and elaborate possible mechanisms of resistance and treatment paths more effective than standard care.   They also discuss the role of MAbs and other drugs besides reviewing particle therapy, T-cell mediated therapies etc. 

The review is written well with 8 different sections and detailed figures.

The authors could discuss the role of T-Cell Transfer Therapy as well. 

First page of introduction:

While describing the radioresistance, the authors could give a mention of these subtypes in Figure 1.

The authors could provide a gist of mitochondrial ROSinduced by Elesclomol in brain tumours. 

The authors could also narrate the role of inhibitors ( nibs), Mibs, Mids etc. as a separate section on drugs

Minor but essential  

In that section, small typos exist.  From "Fortunately" a new sentence could be started 

The tumour microenvironment can be abbreviated as TME

Scores on a scale of 0-5 with 5 being the best 

Language: 4

Novelty: 3

Brevity: 4

Scope and relevance: 4

Comments on the Quality of English Language

 A lot of text has possessive pronouns typed inaccurately.  for example,  "its'"  could be their or could be rewritten  without  "s aphostrope" 

Author Response

Dear reviewer,

Thank you for your detailed and constructive feedback on our work. We appreciate your suggestions and would like to address.

Regarding the inclusion of T-Cell Transfer Therapy,  mitochondrial ROS induced by Elesclomol, and various inhibitors, while we acknowledge its significance and potential in enhancing treatment outcomes, our paper primarily focuses on the complex mechanisms governing the immuno-radiobiological response in primary and secondary brain tumors. Our aim is to provide a comprehensive overview of the interactions within the tumor microenvironment that contribute to radioresistance and the various approaches to overcome these mechanisms. We believe that a detailed discussion on the proposed points would be beyond the scope of this paper, but it certainly warrants attention in future studies.